# Improving the Reliability of Muscle Tissue Characterization Post-Stroke: A Secondary Statistical Analysis of Echotexture Features

**DOI:** 10.3390/jcm14092902

**Published:** 2025-04-23

**Authors:** Borhan Asadi, Juan Nicolás Cuenca-Zaldívar, Alberto Carcasona-Otal, Pablo Herrero, Diego Lapuente-Hernández

**Affiliations:** 1iHealthy Research Group, Instituto de Investigación Sanitaria (IIS) Aragon, University of Zaragoza, 50009 Zaragoza, Spain; basadi@iisaragon.es (B.A.); acarcasona@unizar.es (A.C.-O.); d.lapuente@unizar.es (D.L.-H.); 2Department of Physiatry and Nursing, Faculty of Health Sciences, University of Zaragoza, 50009 Zaragoza, Spain; 3Grupo de Investigación en Fisioterapia y Dolor, Departamento de Enfermería y Fisioterapia, Facultad de Medicina y Ciencias de la Salud, Universidad de Alcalá, 28801 Alcalá de Henares, Spain; nicolas.cuenca@salud.madrid.org; 4Research Group in Nursing and Health Care, Puerta de Hierro Health Research Institute—Segovia de Arana (IDIPHISA), 28222 Majadahonda, Spain; 5Interdisciplinary Group on Musculoskeletal Disorders, Faculty of Sport Sciences, Universidad Europea de Madrid, 28670 Villaviciosa de Odón, Spain; 6Primary Health Center “El Abajón”, 28231 Las Rozas de Madrid, Spain

**Keywords:** ultrasonography, echotexture analysis, muscle tissue, stroke, logistic models

## Abstract

**Background/Objectives:** Ultrasound (US) imaging and echotexture analysis are emerging techniques for assessing muscle tissue quality in the post-stroke population. Clinical studies suggest that echovariation (EV) and echointensity (EI) serve as objective indicators of muscle impairment, although methodological limitations hinder their clinical translation. This secondary analysis aimed to refine the assessment of echotexture by using robust statistical techniques. **Methods:** A total of 130 regions of interest (ROIs) extracted from the gastrocnemius medialis of 22 post-stroke individuals were analyzed. First, inter-examiner reliability between two physiotherapists was assessed by using Cohen’s kappa for muscle impairment classification (low/high) for each echotexture feature. For each examiner, the correlation between the classification of the degree of impairment and the modified Heckmatt scale for each feature was analyzed. The dataset was then reduced to 44 ROIs (one image per leg per patient) and assessed by three physiotherapists to analyze inter-examiner reliability by using Light´s kappa and correlation between both assessment methods globally. Statistical differences in 21 echotexture features were evaluated according to the degree of muscle impairment. A binary logistic regression model was developed by using features with a Cohen’s kappa value greater than 0.9 as predictors. **Results:** A strong and significant degree of agreement was observed among the three examiners regarding the degree of muscle impairment (Kappa_light_ = 0.85, *p* < 0.001), with nine of the 21 features showing excellent inter-examiner reliability. The correlation between muscle impairment classification with the modified Heckmatt scale was very high and significant both globally and for each echotexture feature. Significant differences (<0.05) were found for EV, EI, dissimilarity, energy, contrast, maximum likelihood, skewness, and the modified Heckmatt scale. Logistic regression highlighted dissimilarity, entropy, EV, Gray-Level Uniformity (GLU), and EI as the main predictors of muscle tissue impairment. The EV and EI models showed high explanatory power (Nagelkerke’s pseudo-R^2^ = 0.74 and 0.76) and robust classification performance (AUC = 94.20% and 95.45%). **Conclusions:** This secondary analysis confirms echotexture analysis as a reliable tool for post-stroke muscle assessment, validating EV and EI as key indicators while identifying dissimilarity, entropy, and GLU as additional relevant features.

## 1. Introduction

In recent years, medical image analysis has become an essential tool in the health sciences to improve diagnostic accuracy, disease monitoring, and treatment assessment [1]. The integration of advanced algorithms, including digital image processing, deep learning, and mathematical modeling [2,3] has dramatically improved the ability to extract clinically relevant information from images. These methods facilitate the simulation and quantification of key image features such as texture, structure, and dynamic behavior of organs and tissues, providing clinicians with highly accurate, data-driven information to enhance pathology prognoses and optimize treatment selection [4]. Medical fields that have benefited the most from advanced imaging techniques include oncology [5] and cardiology [6], which remain the leading causes of morbidity and mortality worldwide.

In line with this trend, musculoskeletal research has also benefited from advanced image-based analysis, mainly through tools such as magnetic resonance imaging (MRI) or computed tomography (CT) [7,8], but more recently also through ultrasound (US) imaging and echotexture analysis [9]. Echotexture analysis involves extracting and processing structural and intensity distribution features from US images by using digital image analysis techniques. The most relevant parameters in this area include (1) first-order statistics, which describe the intensity distribution without considering spatial relationships; (2) Haralick features, derived from the gray-level co-occurrence matrix (GLCM), which quantify texture based on the spatial relationships between pixels [10,11]; and (3) gray-level run-length matrix (GLRLM) features, which assess intensity variability and uniformity within the image [12].

US imaging and echotexture analysis are particularly advantageous due to their non-invasive nature, accessibility, and cost-effectiveness, making them promising tools for assessing muscle tissue quality [9,13]. Their application has been extensively studied in various pathological conditions, such as stroke [14], amyotrophic lateral sclerosis [15,16], myofascial pain syndrome [17], chronic low back pain [18], metastatic breast cancer [19], and musculotendinous injuries [20,21], as well as in healthy individuals for various research purposes [22,23].

In the case of stroke, it is characterized by significant neuromuscular alterations produced by central motor pathway disruption, reduced voluntary activation, and peripheral disuse, which leads to pathophysiological changes such as muscle atrophy, fiber-type changes, and progressive replacement of hypoechoic contractile elements with fibrous and adipose tissue [24,25]. Early studies focused mainly on a combination of structural parameters, such as muscle thickness, pennation angle, and fascicle length [26,27], with other echotexture features, mainly echointensity (EI), which have been shown to correlate with clinical features such as spasticity and gait performance [28,29]. More recently, research interest has expanded to additional echotexture features, such as echovariation (EV), which quantifies tissue heterogeneity and may provide complementary information on muscle architecture and degeneration [14]. This research motivation directly led to the further exploration of echotexture features in the present study.

Despite its promising potential, the clinical application of echotexture analysis requires rigorous methodological validation to ensure its reliability, standardization, and reproducibility before it can be incorporated into routine practice. The aforementioned study in post-stroke subjects [14] identified EV and EI as key objective indicators of muscle tissue deterioration, with EV showing a strong correlation with the modified Heckmatt scale, the current gold standard for the qualitative assessment of muscle tissue in individuals with stroke [30]. However, this initial study had several methodological limitations, including a potential selection bias in the US imaging dataset. Rather than using robust statistical validation, the analysis was based on the subjective opinion of expert physiotherapists on the 10 images with the highest values and 10 images with the lowest values for each echotexture feature from a dataset of 130 images. Given these limitations, thorough methodological validation is required before conclusions can be drawn and echotexture analysis can be implemented in daily clinical practice in the post-stroke population.

Recognizing these limitations, the present study aims to perform a secondary analysis to reduce the subjectivity and bias of the previous study by using more robust statistical methods to refine and validate its methodology [14]. In addition, this study aims to improve our understanding of the relationship between echotexture characteristics and clinical muscle classification in individuals who have suffered a stroke. Ultimately, this work aims to provide quantitative evidence to support the integration of echotexture analysis into clinical decision making for the assessment of muscle tissue.

## 2. Materials and Methods

To facilitate a clear understanding of the work developed in this secondary analysis, only the most relevant methodological information is presented in this section. Full details regarding the materials and methods are available in the original article [14], which this analysis is based upon.

### 2.1. Study Design

This secondary analysis was performed by using data from a previously published observational study of the post-stroke population by Asadi et al. [14]. The original study included 130 US images from 22 participants and is the basis for this secondary analysis. The study followed the STROBE guidelines [31] and the ethical principles described in the Declaration of Helsinki [32]. Ethical approval was granted by the Aragon Ethics Committee (PI24/030). The study was registered on ClinicalTrials.gov (NTC06411587).

### 2.2. Participants and Data Collection

In the original study, a physical medicine and rehabilitation specialist recruited post-stroke individuals at Hospital Clínico Universitario Lozano Blesa between February and April 2024. As this is a secondary analysis of existing data [14], the inclusion and exclusion criteria were predefined in the original study. Participants were eligible for inclusion if they were at least 18 years old, had a confirmed diagnosis of stroke based on CT or MRI, had suffered a stroke at least six months before enrollment, and could walk independently with or without assistive devices, with no additional requirements for specific motor impairment. People were excluded if they had other concurrent neurological disorders, such as ataxia or dystonia, had undergone lower-limb surgery, or had medical problems that could have interfered with data collection.

US images of the gastrocnemius medialis muscle were obtained by using the Butterfly iQ+ portable US system (Butterfly Network, Inc., Burlington, MA, USA) in B-mode, employing a linear transducer with a frequency range of 1 to 10 MHz. A standardized imaging protocol was followed with a gain setting of 50% and a depth range of 5–7 cm. Participants were seated with their knees flexed at a 90-degree angle. The US probe was placed at a point corresponding to 30% of the distance between the medial condyle of the tibia and the medial malleolus, and the angle of the probe was adjusted to obtain an optimal image. Six images were collected from each participant, comprising three different images from each lower extremity, with only one region of interest (ROI) per image, specifically a whole-muscle ROI. The decision to assess the gastrocnemius medialis was determined in the original study design. This muscle has been analyzed in previous studies [28] since it plays a key role in gait mechanics, making it a clinically relevant muscle for assessing post-stroke impairment.

### 2.3. Echotexture Feature Extraction

In this secondary analysis, the researchers analyzed the original database, which contained the selected ROIs from the 130 US images used in the study by Asadi et al. [14]. In the original study, the ROIs were selected by two expert physiotherapists, who included as much muscle tissue surface area as possible in the ROI (whole-muscle ROI). In this secondary analysis, the same textural features as in the original study were extracted directly from the database: (1) first-order histogram-based features (see Appendix A); (2) features derived from the gray-level co-occurrence matrix (GLCM) [10,33], computed by using an orientation of 0 degrees, a gray level of 256, and a pixel distance of 5 (see Appendix A); and (3) features based on the gray-level run-length matrix (GLRLM) [12,34,35], calculated across five standard orientations (0, 45, 90, 90, 135, and 180 degrees) with a gray level of 16. The final texture features were obtained by averaging the run-length matrix data and the extracted parameters across all orientations (see Appendix A).

### 2.4. Ultrasound Imaging Assessment

The original study published by Asadi et al. [14] performed an initial feature selection process to identify the most informative texture parameters for assessing muscle tissue impairment. Two independent expert physiotherapists (D.L.H. and P.H.) evaluated 420 ROIs (20 ROIs for each of the 21 echotexture features, selected from the 10 highest and 10 lowest values). Before the study, both physiotherapists jointly reviewed a set of sample US images to reach a consensus on the assessment criteria and standardize image interpretation procedures. Each image contained only one ROI, as it included the maximum amount of muscle, excluding fascia. Each ROI was classified according to two criteria: the degree of muscle tissue impairment, categorized as low or high loss of muscle tissue echotexture, and the modified Heckmatt scale, a four-point grading system, with higher values indicating greater impairment [36]. However, the original study did not analyze the inter-examiner reliability for the level of impairment (low–high), which was the first analysis performed in this study. After performing this reliability analysis, the correlation between the level of impairment, as assessed by the experts (low–high), and the modified Heckmatt scale was also calculated.

After this initial analysis of the original dataset, which was based on echotexture features, a secondary analysis was performed with only 44 ROIs, using a single US image per limb (whole-muscle ROI) per patient to eliminate redundancy and potential bias from multiple images of the same patient. The images were independently assessed by the first two physiotherapists (D.L.H. and P.H.) and a third expert physiotherapist (C.P.F.), who classified them according to the degree of muscle tissue impairment (low–high) and assigned a score based on the modified Heckmatt scale. To preserve methodological rigor and standardization, this third physiotherapist was trained by the first two examiners. The inter-examiner reliability for the level of impairment and the correlation between muscle tissue impairment and the modified Heckmatt scale was reassessed for this final dataset. With these 44 ROIs, a binary logistic regression model was developed by using the degree of muscle tissue impairment as the dependent variable and the echotexture features with a Cohen’s kappa coefficient greater than 0.9 as explanatory variables.

### 2.5. Statistical Analysis

All statistical analyses were performed by using R software (version 4.1.3; R Foundation for Statistical Computing, Vienna, Austria). A significance level of *p* < 0.05 was used. The Shapiro–Wilk test assessed the normality of quantitative variables, presented as means ± standard deviation (SD), while categorical variables were reported as absolute and relative frequencies.

The inter-examiner reliability for the level of muscle tissue impairment for each feature was assessed with Cohen’s kappa statistic and that among the three examiners with Light’s kappa statistic, with both cases being defined as poor (<0.6), moderate (0.6–0.8), strong (0.8–0.9), or excellent (>0.9) [37]. On the other hand, for each examiner, the correlation between the classification of the degree of impairment and the modified Heckmatt scale both globally and for each feature was analyzed based on the polychoric correlation matrix, defining it as negligible (<0.29), low (0.3–0.49), moderate (0.5–0.69), high (0.70–0.89), or very high (>0.90) [38].

In this secondary analysis, the presence of significant differences in echotexture features according to the degree of muscle tissue impairment was analyzed by using the Mann–Whitney U test, while differences in the modified Heckmatt scale were assessed by using Fisher’s exact test. Regarding the binary regression model, a backward stepwise selection approach was applied to optimize the model, minimizing the Akaike Information Criterion (AIC) while excluding variables with a Variance Inflation Factor (VIF) greater than 5. Due to the correlation between EV and EI, separate models were generated by using each as the primary explanatory variable. From the final model, regression coefficients and odds ratios were estimated with their corresponding 95% confidence intervals (CIs) and significance levels. Model fit was assessed by comparing the deviation of the final model to that of the null model, assessing overall model effectiveness, and applying the Hosmer–Lemeshow goodness-of-fit test. The proportion of variance explained was quantified by using Nagelkerke’s pseudo-R^2^. Additionally, the relative contribution of each explanatory variable to the model was evaluated. To assess the model’s predictive performance, receiver operating characteristic (ROC) curve analysis was performed, calculating sensitivity, specificity, area under the curve (AUC), and overall classification accuracy for each significant variable.

## 3. Results

The sample consisted of 16 (72.73%) males and 6 (27.27%) females, with a mean age of 64.32 ± 13.25 years. The inter-examiner reliability for the original 420 ROIs was excellent and significant for the variables EI, EV, dissimilarity, entropy, GLU, homogeneity, kurtosis, RLU, and RPC (Appendix A). A strong and significant degree of agreement was also observed among the three examiners regarding the level of impairment (Kappa_light_ = 0.85, *p* < 0.001). In addition, the correlation between each examiner’s classification of muscle tissue impairment (low–high) and the modified Heckmatt scale grading was very high and significant in total (Examiner_1_ = 0.98 (0.91, 0.99), Examiner_2_ = 0.98 (0.90, 0.99), and Examiner_3_ = 1 (1, 1)) and for all features (Appendix A).

Of the 44 ROIs selected for this secondary analysis, 21 were classified as having low impairment, while the remaining 23 were classified as having high impairment. Statistically significant differences were observed in the level of muscle impairment for the variables EV, EI, dissimilarity, energy, contrast, maximum probability, skewness, and the modified Heckmatt scale (*p* < 0.05). Specifically, ROIs classified as low impairment had a higher proportion of Heckmatt grades 1 and 2, while those classified as high impairment predominantly exhibited grades 3 and 4. In addition, all features evaluated, except EI, showed higher values in the low-impairment group and lower values in the high-impairment group. In contrast, EI showed opposite results, being significantly higher in the high-impairment group and significantly lower in the low-impairment group (Table 1).

After eliminating variables with a VIF greater than five and selecting the model with the lowest AIC, the final logistic regression model retained dissimilarity, entropy, EV, GLU, and EI as the main predictors of muscle tissue impairment. Additionally, the model including EV and the model including EI had a variance of 39, which is lower than the null model variance of 43, indicating a better model fit. In both models, the overall efficacy was statistically significant (χ^2^(4) = 35.36, *p* < 0.001 for EV; χ^2^(4) = 37.38, *p* < 0.001 for EI). The non-significant Hosmer–Lemeshow test (χ^2^(8) = 4.94, *p* = 0.76 for EV; χ^2^(8) = 4.57, *p* = 0.80 for EI) confirmed an overall satisfactory model fit. Nagelkerke’s pseudo-R^2^ indicated high explanatory power, with values of 0.74 and 0.76 for the EV and EI models, respectively. The model fit plots are shown in Appendix A.

For both models, the plot of the predicted values shows that the residuals do not exceed the band (−2, 2), indicating a good fit of the final model selected and its adequacy (Appendix A). All the predicted values had a Cook’s distance of less than 1, both with respect to the observed values and in terms of leverage, indicating the absence of influential values outside the normal limits. However, some atypical ones were observed (Appendix A). The residuals and their standardized versions did not exceed the band (−2, 2) compared with the fitted values, indicating a good fit of the model and its adequacy (Appendix A). The distribution of the residuals in the Q-Q plot was not normal (Appendix A). Finally, the plot of residuals against leverage indicates the presence of some influential data (Appendix A).

In the model including EV, the odds ratio indicated that for each unit increase in EV and dissimilarity, the probability of low impairment increased by 1.28 and 17.86 times, respectively. Conversely, for each unit increase in entropy, the odds of low impairment decreased significantly, by <0.001 times. In the model including EI, each unit increase in EI was associated with a 0.86-fold decrease in the probability of high impairment, while dissimilarity was associated with a 15.65-fold increase in the probability of high impairment. Of all the predictors, EV and EI contributed the most to their respective models (Table 2).

Both models showed high classification performance. The sensitivity of the EV and EI models was 90.48% in both cases, while the specificity rates were 86.96% and 95.65%, respectively. ROC curve analysis showed a significant AUC of 94.20% (95% CI: 87.39–100%) for the EV model and one of 95.45% (95% CI: 89.54–100%) for the EI model (Figure 1).

Regarding classification accuracy, the correct classification rate for EV was 22.73%, while that for EI was 84.09%. The classification rates for dissimilarity and entropy were 29.55% and 63.64%, respectively.

## 4. Discussion

This secondary analysis builds on the methodological foundations established in the initial exploratory study [14] by incorporating advanced statistical validation and refining the dataset to improve its clinical applicability. The results demonstrate high inter-examiner reliability, reinforcing the robustness of the original classification (low–high impairment) developed from the previous study. Including a third independent examiner further reduced potential bias and strengthened the consistency of the original classification approach. In addition, by ensuring that only one US image per limb was selected from each patient, this study mitigated the potential over-representation of individual patients, which may have influenced the original study results.

The results of this secondary analysis align with those of the original study, confirming that echotexture features, particularly EV and EI, are valuable indicators of muscle tissue impairment in post-stroke individuals. Logistic regression models and ROC curve analyses provided a more rigorous assessment of the predictive power of these features, demonstrating high classification performance with excellent sensitivity and specificity. Importantly, this study also identified additional relevant features, such as dissimilarity, entropy, and GLU, which were not initially considered as key features in the exploratory analysis performed in the original study [14]. These results suggest that traditional expert-based assessment may not fully capture all relevant structural changes in muscle tissue. Although expert judgment remains essential, specific echotexture changes may not be visually apparent but can be objectively quantified by using statistical models and predictive analysis.

Our results confirm the importance of EI and EV first-order echotexture features in assessing muscle tissue quality in US images, as previously highlighted in the original research study [14]. Previous studies have also highlighted the importance of these parameters to classify different types of soleus muscle injuries [20], to differentiate individuals with patellar tendinopathy from healthy controls [21], and to assess gastrocnemius medialis and soleus muscle tissue in plantar fascia pathology [39]. Apart from this use in musculoskeletal conditions, EV has shown good accuracy for muscle tissue assessment in individuals with amyotrophic lateral sclerosis [16,40]. The association between high EI and increased muscle tissue impairment is well supported by the previous literature describing histopathological changes such as disuse-induced atrophy and fibroadipose replacement of hypoechoic contractile elements [41,42]. In parallel, lower EV values observed in more impaired muscles may reflect the loss of structural complexity and tissue differentiation, as fibrous and adipose infiltration replace the heterogeneous architecture of healthy muscle tissue. However, while first-order features remain valuable, they may not be sufficient for more complex image processing tasks, where higher-order texture parameters, such as those derived from GLCM and GLRLM, offer superior diagnostic accuracy [9].

Building on this premise, our secondary analysis highlights that in addition to EI and EV, some GLCM (dissimilarity and entropy) and GLRLM (GLU) features may be key indicators of muscle tissue integrity in post-stroke populations, although the original research study did not identify them as primary indicators [14]. This is consistent with prior research demonstrating that GLCM and GLRLM features improve the accuracy of supraspinatus tendon rupture classification [43] and are also relevant in characterizing forearm flexor and quadriceps muscle tissue in individuals with amyotrophic lateral sclerosis [16]. Notably, the latter study in amyotrophic lateral sclerosis suggests that the integration of echotexture features with conventional muscle US variables, such as muscle thickness, may further improve diagnostic capabilities [16]. In light of these data, future research should explore comprehensive models that integrate first-order, GLCM, and GLRLM features alongside conventional US parameters to refine the accuracy and clinical applicability of muscle echotexture analysis.

A fundamental difference between this secondary analysis and the original study by Asadi et al. [14] is the image selection method. The original dataset included three US images per limb for each participant, which, while helpful in assessing measurement reliability, may have introduced redundancy and potential bias in statistical comparisons. In contrast, this secondary analysis refined the dataset by selecting only one representative image per limb for each patient, aligning with standard clinical practice and ensuring a more rigorous assessment of muscle tissue impairment.

This study also addresses one of the limitations identified in the original research work [14], which is inter-examiner reliability. By including additional examiners and using Cohen’s and Light’s kappa to assess agreement, the present study demonstrates strong reliability of echotexture analysis for assessing muscle tissue. However, some limitations remain, such as the relatively small sample size and the inability to conduct demographic subgroup analysis (e.g., sex, age, and time since stroke onset) due to this constraint, which could influence the results and limit their generalizability. In addition, although this study employed the rectangular ROI technique (also named the square technique) to select the amount of muscle tissue area for image analysis, as performed in previous studies [16,40], there are alternative approaches, such as the trace technique, in which the classifier delineates all visible muscle without artefacts. Given these differences, it is essential to emphasize that echotexture outcomes may not be interchangeable between the rectangular/square and trace techniques [44], mainly because the use of irregular or manually traced ROIs may introduce computational challenges by not conforming to standard matrix-based texture analysis frameworks.

These results further support the integration of echotexture analysis as a quantitative tool to assess muscle tissue quality in post-stroke individuals. In contrast to traditional qualitative assessments, echotexture features provide objective and reproducible metrics that could improve clinical decision making. However, future studies should validate these results in larger cohorts to confirm the reliability of echotexture features across various demographic groups and further explore their correlation with functional outcomes or patient-reported symptoms. Further studies are needed to determine their clinical relevance for monitoring treatment response and therapeutic progress, enabling necessary adjustments to intervention. In addition, future research should explore the usefulness and predictive ability of echotexture analysis on long-term rehabilitation outcomes, as echotexture features may help clinicians to prevent the adverse effects of some treatments (such as botulinum toxin) and to tailor and personalize treatment for each patient [45], as well as the long-term prognostic value of echotexture features in predicting muscle tissue impairment post-stroke. Furthermore, further research into echotexture image locations is needed, as an existing study found significant differences in texture analysis between hamstring muscles and measurement locations [23], suggesting that the assessment of different muscle locations may influence echotexture values. This finding underscores the importance of standardizing image acquisition protocols for future research. Finally, it would be interesting for future studies to explore texture analysis approaches adapted to non-rectangular ROI selections.

## 5. Conclusions

This secondary analysis reinforces the validity of echotexture analysis for assessing muscle tissue quality in post-stroke individuals by integrating advanced statistical techniques and refining the methodological limitations present in an original exploratory study. The findings support the clinical utility of echotexture analysis as a quantitative assessment tool, confirming EV and EI as the most reliable indicators of muscle tissue impairment while also identifying other potential echotexture features from GLCM and GLRLM, such as dissimilarity, entropy, and GLU. However, further research is needed to establish standardized imaging protocols, validate these findings in larger cohorts, and investigate the potential of echotexture features to inform rehabilitation strategies and therapeutic interventions.

## Figures and Tables

**Figure 1 jcm-14-02902-f001:**
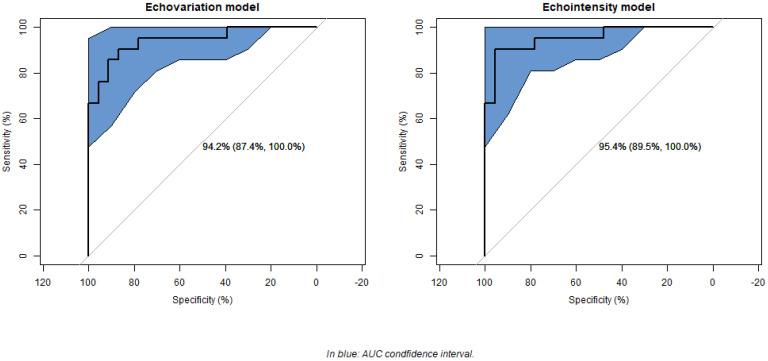
Model receiver operating characteristic (ROC) curves.

**Table 1 jcm-14-02902-t001:** Descriptive statistics and comparison of echotexture features between ROIs classified as low and high impairment after the secondary analysis.

		Overall	Low Impairment	High Impairment	^a^ *p*-Value
N		44	21	23	
**Main outcomes**					
Echovariation		46.26 ± 16.69	56.84 ± 16.52	36.60 ± 9.58	<0.001 *
Echointensity		82.89 ± 23.60	66.56 ± 17.92	97.80 ± 17.58	<0.001 *
**Secondary outcomes**					
Heckmatt scale, *n* (%)	Grade 1	3 (6.8)	3 (14.3)	0 (0.0)	<0.001 *
	Grade 2	21 (47.7)	18 (85.7)	3 (13.0)	
	Grade 3	12 (27.3)	0 (0.0)	12 (52.2)	
	Grade 4	8 (18.2)	0 (0.0)	8 (34.8)	
Variance		1237.73 ± 324.37	1267.46 ± 301.08	1210.59 ± 348.76	0.568
Standard deviation		34.87 ± 4.71	35.34 ± 4.42	34.45 ± 5.02	0.536
Skewness		0.50 ± 0.34	0.65 ± 0.32	0.37 ± 0.31	0.005 *
Kurtosis		0.03 ± 0.65	0.22 ± 0.81	−0.15 ± 0.40	0.056
Correlation		0.97 ± 0.01	0.97 ± 0.01	0.98 ± 0.01	0.103
Dissimilarity		5.97 ± 0.74	6.21 ± 0.81	5.75 ± 0.60	0.037 *
Energy		0.02 ± 0.01	0.03 ± 0.02	0.02 ± 0.00	0.042 *
Contrast		62.00 ± 14.82	68.20 ± 15.69	56.35 ± 11.65	0.007 *
Homogeneity		0.17 ± 0.03	0.18 ± 0.04	0.17 ± 0.02	0.444
Angular Second Moment		0.00 ± 0.00	0.00 ± 0.00	0.00 ± 0.00	0.058
Maximum probability		0.01 ± 0.02	0.01 ± 0.02	0.00 ± 0.00	0.013 *
Entropy		7.04 ± 0.22	7.00 ± 0.24	7.07 ± 0.20	0.275
Cluster Shade		8.03 ± 0.23	7.99 ± 0.25	8.07 ± 0.20	0.234
Cluster Prominence		10.87 ± 0.36	10.81 ± 0.42	10.92 ± 0.28	0.294
Short-Run Emphasis		0.73 ± 0.08	0.73 ± 0.07	0.72 ± 0.08	0.49
Long-Run Emphasis		440.64 ± 134.70	447.98 ± 173.53	433.94 ± 89.38	0.734
Gray-Level Uniformity		10,957.52 ± 2970.64	10,237.13 ± 2319.04	11,615.27 ± 3378.09	0.126
Run-Length Uniformity		17,268.72 ± 4566.54	17,109.75 ± 2938.33	17,413.87 ± 5732.68	0.828
Run Percentage		15.27 ± 3.28	15.37 ± 2.45	15.17 ± 3.95	0.837

Data are expressed as means ± standard deviation or as absolute and relative values (%). ^a^ significant if *p* < 0.05 (marked with asterisk).

**Table 2 jcm-14-02902-t002:** Final model summary.

	Odds Ratio (95% CI)	Coefficient (SE)	95% CI	Z-Value (^a^ *p*-Value)	Variable Importance
**Echovariation main outcome model**					
(Intercept)	7.77 × 10^31^ (111,740.12, 1.63 × 10^73^)	73.43 (SE = 38.61)	11.62, 168.57	0.057	
Echovariation	1.275 (1.1, 1.62)	0.24 (SE = 0.09)	0.09, 0.48	0.012 *	2.52
Dissimilarity	17.856 (2.61, 360)	2.88 (SE = 1.19)	0.96, 5.88	0.016 *	2.40
Entropy	<0.001 (0.001, 0.01)	−13.75 (SE = 6.31)	−29.66, −3.98	0.029 *	2.17
Gray-Level Uniformity	1 (0.99, 1)	0 (SE = 0)	−0.001, 0	0.154	1.42
**Echointensity main outcome model**					
(Intercept)	3.64 × 10^20^ (0, 5.00 × 10^54^)	47.34 (SE = 32.57)	−8.6, 125.95	0.146	
Echointensity	0.86 (0.74, 0.93)	−0.14 (SE = 0.05)	−0.29, −0.06	0.009 *	2.62
Dissimilarity	15.65 (2.33, 287.63)	2.75 (SE = 1.17)	0.84, 5.66	0.019 *	2.33
Entropy	0.001 (0, 5.02)	−6.69 (SE = 4.87)	−18.43, 1.61	0.170	1.37
Gray-Level Uniformity	1 (0.99, 1)	0 (SE = 0)	−0.001, 0	0.142	1.46

95% CI: 95% confidence interval; SE: Standard Error. ^a^ significant if *p* < 0.05 (marked with asterisk).

## Data Availability

The data presented in this study are available from the corresponding author upon request, due to the fact that these data are part of a larger, ongoing project and may not yet be fully available for public sharing until the project is completed.

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
