# Peer review of "Improving the Reliability of Muscle Tissue Characterization Post-Stroke: A Secondary Statistical Analysis of Echotexture Features"

_jcm, 2025, doi:10.3390/jcm14092902_

Round 1
Reviewer 1 Report
Comments and Suggestions for Authors
1.The study includes a relatively small sample size (n=22 participants) with 44 regions of interest (ROIs). While the statistical analysis is robust, the limited sample may restrict the generalizability of findings to broader post-stroke populations. Future studies should validate these results in larger cohorts to confirm the reliability of echotexture features across diverse demographics.
2.The decision to reduce the dataset to one ROI per limb per patient addresses redundancy but raises questions about whether a single image sufficiently captures muscle heterogeneity. A justification for this selection criterion or a sensitivity analysis comparing single vs. multiple ROIs would strengthen the methodology.
3.While Cohen’s kappa demonstrated excellent reliability among examiners, the manuscript lacks details on standardization protocols (e.g., training, calibration) for ultrasound image acquisition and ROI selection. Clarifying these protocols would enhance reproducibility and clinical adoption.
4.The study does not explore potential confounding effects of age, sex, or time since stroke onset on echotexture features. Subgroup analyses could elucidate whether these variables influence muscle impairment classification, refining clinical applicability.
5.The Butterfly iQ+ system settings (e.g., gain, depth) are described, but additional details (e.g., probe frequency, spatial resolution) are omitted. Full technical specifications are critical for replication, especially given the sensitivity of texture analysis to imaging parameters.
6.While the logistic regression models show high predictive performance (AUC >94%), the clinical relevance of these models (e.g., thresholds for intervention, correlation with functional outcomes) remains underexplored. Linking echotexture features to rehabilitation efficacy or patient-centered outcomes would bolster clinical significance.
7.The inverse relationship between EI and muscle impairment (higher EI in high impairment) conflicts with prior literature (e.g., EI increases in fatty infiltration). A discussion reconciling these findings with existing mechanistic models of muscle degeneration would clarify the physiological basis of EI and EV in stroke.
8.The study uses a rectangular ROI method, but alternative approaches (e.g., trace technique) may yield different echotexture values. A brief comparison of ROI selection methods in the limitations section would guide future research on protocol harmonization.
Comments on the Quality of English LanguageThe English could be improved to more clearly express the research.
Author Response
I attach the reply point by point to all the reviewers

Reviewer 2 Report
Comments and Suggestions for Authors
For the authors
This study conducts a secondary statistical analysis to improve the reliability of muscle tissue characterization post-stroke using echotexture features. A total of 130 regions of interest (ROIs) from 22 stroke patients were analyzed with robust statistical techniques. The results confirm that echovariation (EV) and echointensity (EI) are key indicators of muscle impairment
A lot of effort has been expended, while I suggest that the author clarify the following concerns.
- The Introduction section should provide a detailed description of US indicators related to muscle impairment assessment, citing relevant literature to describe the progress in this area. Begin with a brief paragraph describing the pathophysiology of “stroke and muscle tissue deterioration” if this is the primary assessment focus. The rest paragraph is not so necessary.
- The authors state that "Participants were eligible for inclusion if they were at least 18 years old, had a confirmed diagnosis of stroke based on CT or MRI, had suffered a stroke at least six months before enrolment, and could walk independently with or without assistive devices." It is unclear whether the intention was to include patients with limb motor dysfunction or simply stroke survivors. If the latter, should damage to brain motor areas (e.g., the precentral gyrus) be excluded? Please redefine the inclusion and exclusion criteria.
- Why was the gastrocnemius medialis muscle specifically chosen for this study? Were there any particular clinical or anatomical reasons for focusing on this muscle, especially considering the diverse range of muscle impairments that can occur post-stroke?
- How was the severity of muscle tissue deterioration assessed in the included patients? Were any specific scales or diagnostic tools used to quantify the extent of muscle impairment? How does muscle tissue deterioration, as assessed by echotexture features, correlate with functional outcomes or patient-reported symptoms? Were any functional assessments conducted alongside the echotexture analysis?
- Please condense the "Statistical Analysis" section to highlight only the statistical methods relevant to the primary outcomes.
- The study reports excellent inter-examiner reliability for several echotexture features using Cohen's kappa. However, the reliability analysis was performed on 420 ROIs initially and then reduced to 44 ROIs for the secondary analysis. How does the reduction in the number of ROIs affect the reliability estimates? Were any additional reliability tests conducted on the final dataset of 44 ROIs?
- Please provide detailed definitions and calculations for Echovariation (EV) and Echointensity (EI) in the study, including specific formulas or methods used to derive these features from ultrasound images. Additionally, discuss the biological basis for changes in EV and EI in post-stroke muscle tissue and their correlation with underlying pathological processes such as muscle atrophy, fibrosis, or fatty infiltration.
- How were the 130 ROIs distributed among the 22 patients? Specifically, how many ROIs were selected per patient, and what criteria were used to determine the number of ROIs per muscle? Was there a consistent number of ROIs per patient, or did it vary?
- What is the long-term prognostic value of echotexture features in predicting muscle tissue deterioration post-stroke? I suggest a table to evaluate this is needed.
Reviewer.
Author Response

(The authors gave the same response as above.)

Reviewer 3 Report
Comments and Suggestions for Authors
The study's methodological improvements deserve particular praise:
1. The reduction from 130 to 44 ROIs (one image per limb per patient) effectively eliminated redundancy and potential bias from multiple images of the same patient
2. Adding a third independent examiner strengthened reliability assessments
3. The application of advanced statistical techniques (binary logistic regression, ROC analysis) provided more rigorous validation than the subjective assessment used in the original study
4. The comparison with the modified Heckmatt scale offered valuable clinical validation
Significance and Implications
This research makes several important contributions to post-stroke muscle assessment:
1. It confirms echotexture analysis as a reliable quantitative tool for assessing muscle tissue quality
2. It validates previously identified parameters (EV and EI) while identifying additional relevant features (dissimilarity, entropy, GLU)
3. It provides a more objective alternative to traditional qualitative assessment methods
4. The demonstrated high classification accuracy suggests potential clinical utility for diagnosis and monitoring
Limitations and Future Directions
Despite its strengths, the study has some limitations the authors acknowledge:
1. The relatively small sample size (22 patients) limits generalizability
2. The lack of demographic subgroup analysis prevents understanding of how factors like age, sex, and time since stroke might influence results
3. The rectangular ROI technique used may not be interchangeable with other approaches like the trace technique
Future research should focus on standardizing imaging protocols, validating findings in larger cohorts, and exploring the predictive capacity of echotexture features for rehabilitation outcomes and treatment response monitoring.
Author Response

(The authors gave the same response as above.)

Round 2
Reviewer 1 Report
Comments and Suggestions for Authors
The authors have addressed the methodological concerns raised in the initial review through rigorous revisions, significantly enhancing the study's reliability and clinical relevance. By refining the dataset to eliminate redundancy (reducing from 130 to 44 ROIs), incorporating robust statistical validation (e.g., Cohen’s kappa for inter-examiner reliability), and introducing a third independent examiner to reduce bias, the revised analysis demonstrates improved rigor and reproducibility. The logistic regression models and ROC analyses robustly validate EV and EI as key predictors of muscle impairment, while identifying additional GLCM/GLRLM features (dissimilarity, entropy, GLU) as clinically relevant. Results are well-supported by statistical evidence, with high sensitivity (90.48%) and specificity (86.96–95.65%), and the discussion contextualizes findings effectively against prior literature. Limitations, such as sample size constraints and the need for standardized protocols, are appropriately acknowledged. Minor revisions, such as clarifying the impact of sample size on generalizability or detailing examiner calibration procedures, could further strengthen the manuscript.
Author Response
I attach reply

Reviewer 2 Report
Comments and Suggestions for Authors
For the authors
The authors have provided excellent responses, and I believe most of the issues have been clarified. Thank you for your efforts. However, further clarification is still needed from the authors regarding the following questions:
- I notice that the authors have added background information in the Introduction (page 2, lines 76-86), which provides motivation for the study. Given that significant pathological changes occur in muscle tissue post-stroke, there is a need to identify tools that can accurately assess these changes. Early studies primarily focused on structural parameters, such as muscle thickness, pennation angle, and fascicle length, as well as echointensity (EI), which laid the foundation for subsequent research. More recently, research interest has expanded to additional echotexture features, such as echovariation (EV), which may offer complementary information on muscle architecture and degeneration. I suggest the authors add a concluding sentence to this paragraph, to explain “This research motivation directly led to the further exploration of echotexture features in the present study.”
- In response 2 and 3, the authors explained the methods, stating that this study is a secondary analysis of the data from a previous study (reference 14). I noticed that the authors have addressed this on page 3, lines 109-112. However, this is insufficient because many readers may still be confused. I suggest explicitly stating in this paragraph: "This study is a secondary analysis of an original data [14], and inclusion and exclusion criteria were established," and appropriately clarifying the following in the Methods section:
- "The original study details that participants were included based on a confirmed diagnosis of stroke and the ability to walk independently, with no additional requirements for specific motor impairments. The only neurologic exclusion criterion was the presence of other concurrent neurologic conditions."
- "The decision to assess the medial gastrocnemius was determined in the original study design. This muscle has been analyzed in previous studies since it plays a key role in gait mechanics, making it a clinically relevant muscle for assessing post-stroke impairments."
- Please move the supplementary table (Table S1) to the main body of the manuscript.
Reviewer.
Author Response
I attach reply
